# FEM Simulation-Based Adversarial Domain Adaptation for Fatigue Crack Detection Using Lamb Wave

**DOI:** 10.3390/s23041943

**Published:** 2023-02-09

**Authors:** Li Wang, Guoqiang Liu, Chao Zhang, Yu Yang, Jinhao Qiu

**Affiliations:** 1State Key Laboratory of Mechanics and Control of Mechanical Structures, Nanjing University of Aeronautics and Astronautics, Nanjing 210016, China; 2Structural Damage Monitoring Laboratory, Aircraft Strength Research Institute of China, Xi’an 710065, China

**Keywords:** fatigue crack detection, lamb waves, finite element method, domain-adversarial neural network, maximum mean discrepancy, metal structures

## Abstract

Lamb wave-based damage detection technology shows great potential for structural integrity assessment. However, conventional damage features based damage detection methods and data-driven intelligent damage detection methods highly rely on expert knowledge and sufficient labeled data for training, for which collecting is usually expensive and time-consuming. Therefore, this paper proposes an automated fatigue crack detection method using Lamb wave based on finite element method (FEM) and adversarial domain adaptation. FEM-simulation was used to obtain simulated response signals under various conditions to solve the problem of the insufficient labeled data in practice. Due to the distribution discrepancy between simulated signals and experimental signals, the detection performance of classifier just trained with simulated signals will drop sharply on the experimental signals. Then, Domain-adversarial neural network (DANN) with maximum mean discrepancy (MMD) was used to achieve discriminative and domain-invariant feature extraction between simulation source domain and experiment target domain, and the unlabeled experimental signals samples will be accurately classified. The proposed method is validated by fatigue tests on center-hole metal specimens. The results show that the proposed method presents superior detection ability compared to other methods and can be used as an effective tool for cross-domain damage detection.

## 1. Introduction

As a unique non-destructive evaluation (NDE) system, structural health monitoring (SHM) has shown great potential in reducing maintenance cost, extending service life, and ensuring structural integrity [1,2]. Several SHM techniques have been implemented for damage detection in the past few years, such as strain-based SHM [3], electromechanical impedance-based SHM [4], smart coating-based SHM [5], Lamb waves-based SHM [6], etc. Due to the long-distance propagation and small attenuation of Lamb waves in structures, Lamb waves-based damage detection technology has received extensive concerns.

However, the complexities involved with Lamb wave due to its multi-modal and dispersive nature make the signals analysis quite strenuous [7], and its physics modeling to predict the output and identifying the damage [8,9] is a difficult and prohibitive task. Conventional Lamb wave-based damage detection methods are to extract predesigned damage features of Lamb wave in time and frequency domain and identify structural damages by comparing damage features with their thresholds [10,11]. Due to the effect of structures geometry on Lamb wave, the damage feature threshold needs to be adjusted according to different structures, which often presents a less robustness and poor knowledge generalization performance in real life structures with complicated geometry. These methods also need a reasonable selection of damage features, which highly rely on expert experience. To circumvent these limitations, many damage detection methods based on machine learning have been developed for automatic damage detection without a specific threshold. Atashipour et al. [12] proposed an automatic damage identification approach for steel beams based on Lamb wave and artificial neural network (ANN). Damage character points based on continuous wavelet transform were extracted first, then a multilayer ANN supervised by error-back propagation algorithm was trained to automatic detect damage. Li et al. [13] used Hilbert transform, power spectral density, fast Fourier transform, and wavelet fractal dimension to extract multi-features from time domain, frequency domain, and fractal dimension of Lamb wave. Following that, a machine learning method based on support vector machine (SVM) was used to fuse multi-features and further identify damage. Yang et al. [14] developed an integrated damage identification method based on least margin, which integrates multiple machine-learning models and outputs the fused damage identification result by polling all models’ decisions. Twelve damage features and seven machine learning methods, including k-nearest neighbor (KNN), radial basis function support vector machine (RBF-SVM), Gaussian process (GP), decision tree (DTree), neural network (NN), Gaussian naive Bayes (GNB) and quadratic discriminant analysis (QDA), were applied to predict the damage identification results.

Instead of manual feature extraction for machine learning, various damage detection methods based on deep learning have been developed to automatically feature extraction and damage detection. Lee et al. [15] adopted a deep autoencoder (DAE) to capture hidden representation and effective tracking of signal variations, and the reconstruction error was used to diagnosis fatigue damage in composites structures. Chen et al. [16] and Wu et al. [17] converted the Lamb wave signals into a two-dimensional time-frequency spectrogram with the continuous wavelet transform, then input them into a 2D convolutional neural network (CNN) to classify damage. To minimize loss of information in conversion of time signals to image, Pandey et al. [7] used 1D CNN to detect damage directly using original Lamb wave signals of an aluminum plate, in which Lamb wave response signals obtained with FEM simulations were used as training samples and experimental data were used for testing. Sampath et al. [18] incorporated the long short-term memory (LSTM) with trispectrum-based higher-order spectral analysis to propose a novel hybrid method for reliable fatigue crack detection under noisy environments. The DL model based on LSTM was used to eliminate the random noise by reconstructing the original Lamb wave signals, and trispectrum-based higher-order spectral analysis method was adopted to extract the nonlinear components considered as an indication of fatigue cracks. Yang et al. [19] used the temporal distributed conventional neural network (TDCNN) to extract less expertise-dependent features, in which the long short-term memory (LSTM) was used to associate features of data fragments.

However, these data-driven intelligent damage detection methods require sufficient labeled data to train the intelligent model for good performance, and the training and testing data must follow the same distribution. In industrial scenarios, collecting sufficient labeled data for training is usually expensive and impractical, and usually just a large amount of unlabeled data can be obtained. Meanwhile, many available training data are obtained by simulations or from simple structure forms, which may not follow the same distribution with data from the practical complex conditions. When the training data are non-labeled or the distributions are mismatched, the performance of data-driven intelligent damage detection methods may drop sharply.

Transfer learning is an effective knowledge generalization tool and can transfer the knowledge learned from an abundant labeled source domain into a new but related target domain [20]. By combining the hidden features of learning ability and deep learning and knowledge transfer ability of transfer learning, deep transfer learning has been widely studied. Nowadays, deep domain adaptation has a dominant position in deep transfer learning, which can be summarized into three categories: discrepancy-based, adversarial discrimination-based, and adversarial generation-based deep domain adaptation methods [21]. The main idea of discrepancy-based deep domain adaptation methods is to minimize the distribution discrepancy between source domain and target domain to get domain-invariant features, in which MMD [22,23,24], Kullback–Leibler (KL) divergence [25,26,27], and Wasserstein distance [28,29], et al., can be used as the metrics of inter-domain distribution divergence. Deep domain confusion (DDC) [23] and deep adaptation network (DAN) [24] are the classical discrepancy-based deep domain adaptation methods. Adversarial discrimination-based deep domain adaptation methods aim to extract fault-discriminative and domain-invariant features through adversarial training with the gradient reversal layer (GRL) [30]. DANN [31] is a classical adversarial discrimination-based deep domain adaptation method of feed-forward architectures, combining domain adaptation and deep feature learning within one training process. Adversarial generation-based deep domain adaptation methods aim to minimize the discrepancy between the source data and target data with the adversarial training. The widely used adversarial generation-based deep domain adaptation models include generation adversarial network (GAN) [32], Wasserstein-GAN (WGAN) [33], Wasserstein-GAN with gradient penalty (WGAN-GP) [34], et al. Many deep domain adaptation models for structural health monitoring using Lamb wave have been developed. Alguri et al. [35] proposed a transfer learning framework for Lamb waves’ full wave field reconstruction, in which autoencoder was used to learn the general propagation of signals, then the learned knowledge was combined with sparse spatial measurements to reconstruct full wavefield. Zhang et al. [36] applied the joint distribution adaptation (JDA) to adapt both the marginal distribution and conditional distribution of the Lamb waves from aluminum plate and composite plate, then used the LSTM network to learn the damage indexes for damage probability imaging. Zhang et al. [37] proposed a muti-task deep transfer learning methods by transferring the high-level shared features of damage level detection task to damage location task. Wang et al. [38] used the MMD-based deep adaptation network for learning transferable features to make the classifier trained on the labeled source data achieve comparable performance on the unlabeled target data. Even though several deep transfer learning methods have been developed for Lamb wave-based damage monitoring, fatigue crack detection with deep transfer learning model for Lamb wave has not yet been fully studied.

In order to automatically detect fatigue crack and further improve the detection accuracy, in this paper, a deep domain adaptation method based on FEM simulation and MMD-DANN was proposed for damage detection using Lamb wave. FEM simulations are employed to obtain the simulated response signals under different conditions. Therefore, the insufficient labeled data of real-world can be possibly solved. However, the variabilities of real response signals during the fatigue crack growth have not been represented in the simulated response signals. For engineering structures, the real response signals are affected by complex uncertainties, like structure manufacturing procedure, environmental variables, crack geometries, multi-sensors performance, and the sensor installation process. Thus, the classifier model of simulated signals cannot be directly applied to the real structures. By fusing the distribution discrepancy metric and the adversarial discrimination training to minimize the domain disparity of the simulated source data and experimental target data, the MMD-DANN model was developed to learn damage-discriminative and domain-invariant feature representations. Then, the classifier model of simulated signals can be directly transferred to experimental signals with comparative ability.

The outline of this paper is as follows. The details of the proposed fatigue crack detection method are specified in Section 2, including the network architecture of MMD-DANN model, training of MMD-DANN model, and detection procedure of target domain. In Section 3, the proposed method is demonstrated through fatigue test data of center-hole metal specimens. Conclusions are drawn in Section 4.

## 2. Proposed MMD-DANN-Based Fatigue Crack Detection Method

By combining the distribution discrepancy metric of MMD and the adversarial discrimination training of DANN model, an unsupervised deep domain adaption method based on MMD-DANN model was proposed to detect fatigue crack in this paper, which bridges the source and target domains in an isomorphic latent feature space and performs a superior diagnosis performance on the unlabeled target data.

In the proposed method, the simulated Lamb wave response signals obtained from the undamaged case and multiple damaged cases are assigned as the labeled source domain data. The experimental Lamb wave response signals obtained from the fatigue test are assigned as unlabeled target domain data. Assume that the source and target domain data are 𝒟*_s_* = {$x_{s}^{i}$, $y_{s}^{i}$${\}}_{i=1}^{m}$ and 𝒟*_t_* ={$x_{t}^{j}$${\}}_{j=1}^{n}$, in which 𝒟*_s_* are the *m* labeled source samples and 𝒟*_t_* are the *n* unlabeled target samples. ***x****_s_*, *y_s_* ∈ {1, …, *K*} are the simulated response signals and the corresponding labels for *K* types of damage categories in the source domain.***x**_t_* are the unlabeled experimental response signals in the target domain.

### 2.1. Network Architecture of MMD-DANN Model

The architecture of the proposed MMD-DANN model is illustrated in Figure 1, with all hyperparameters used in the paper given besides the layers. It consists of a deep feature extractor ***G****_f_*, a deep label predictor ***G****_y_*, and a domain classifier ***G****_d_*. As shown in Figure 1, the labeled source and unlabeled target Lamb wave signals are first input into the feature extractor ***G****_f_* to extract the multi-dimensional features vector. The label predictor ***G****_y_* takes the extracted features of the labeled source data as input and predicts the class labels. The domain classifier ***G****_d_* takes the extracted features of the source and target data as input and predicts the domain labels. The distribution discrepancy of the extracted features is reduced by multi-layer domain adaptation and the adversarial training between the feature extractor and domain classifier.

The feature extractor ***G****_f_* is composed of three 1-D convolutional layers (CLs) and one flattened layer. The features of the input response signals are extracted through three layers of convolutional computation, and the resulted features map is compressed down to multi-dimensional features vector through a flattened layer. The rectified linear unit (ReLU) was used as an activation function of every CL to improve computational efficiency [39], and a batch normalization (BN) layer was used after every CL to normalize the data and reduce the internal covariate shift [40]. The weights of ***G****_f_* of the source and target domain are shared.

The label predictor ***G****_y_* is made up of three fully connected layers (FCLs). The high-dimensional features vector of the source domain extracted by ***G****_f_* is input to ***G****_y_* and further compressed through two FCLs with ReLU activation function. The sigmoid activation function is used in the third FCL to predict the class label. The weights of ***G****_y_* of the source and target domain are also shared.

The domain classifier ***G****_d_* is composed of two FCLs. The high-dimensional features vector of the source and target domain extracted by ***G****_f_* are used as input and further compressed through one FCL with ReLU activation function. The softmax function is used in the second FCL to predict the domain label. The weights of ***G****_d_* of the source and target domain are also shared.

### 2.2. Training of MMD-DANN Model

According to the three outputs of MMD-DANN model, three losses are constructed, including the label prediction loss *L_y_* based on the outputs of the label predictor ***G****_y_*, the domain classification loss *L_d_* based on the outputs of domain classifier ***G****_d_*, and the domain adaptation loss *L_MMD_* based on the outputs of the feature extractor ***G****_f_*, as illustrated in Figure 1. Three losses are detailed as follows:

The label predictor ***G****_y_* takes the extracted features of the labeled source data as input and outputs the predicted labels. During the training stage, for the good performance of the label predictor, the label predictor ***G****_y_* aims to minimize the label prediction loss between the predicted label and the actual label for the training source data in a supervised way. The parameters *θ_f_*, *θ_y_* of both the feature extractor ***G****_f_* and the label predictor ***G****_y_* are optimized at the same time. The label prediction loss *L_y_* of the label predictor ***G****_y_* based on the cross-entropy loss function is defined as:(1)$$L_{y}\left(D_{s}\right)=-E_{\left(x_{s},y_{s}\right)∼D_{s}}\sum_{k=1}^{K}1_{\left[k=y_{s}\right]}\log\left(G_{y}\left(G_{f}\left(x_{s}\right)\right)\right)$$
where E_(***x***_*_s_*_,_ *_y_*_s)~𝒟_*_s_* means calculates the expectation of the samples from 𝒟*_s_*. 1_[*K* = _*_ys_*_]_ is an indicator function, if *K* = *y_s_*, its value equals 1, else it equals 0.

Simultaneously, the domain classifier ***G****_d_* takes the extracted features of the source and target domain data as input and outputs the predicted domain labels. During the training stage, for the good performance of the domain classifier, the domain classifier ***G****_d_* aims to minimize the domain classification loss of two domains in a supervised way. The domain label of the source domain data is assigned as 0, and the domain label of the target domain data is assigned as 1. The parameters *θ_f_*, *θ_d_* of both the feature extractor ***G****_f_* and the domain classifier ***G****_d_* are optimized at the same time. The domain classification loss *L_d_* of the domain classifier ***G****_d_* based on the cross-entropy loss function is defined as:(2)$$L_{d}\left(D_{s},D_{t}\right)=-E_{x_{s}∼D_{s}}\left[\log\left(G_{d}\left(G_{f}\left(x_{s}\right)\right)\right)\right]-E_{x_{t}∼D_{t}}\left[\log\left(1-G_{d}\left(G_{f}\left(x_{t}\right)\right)\right)\right]$$

To further reduce the distribution discrepancy between the two domains and improve the domain adaptation of the feature extractor, the distribution discrepancy of the output features vector between two domains is measured and incorporated into the model training, which is defined as the domain adaptation loss *L_MMD_*. MMD is a common distance metric in deep domain adaptation to measure the distribution discrepancy between two datasets. MMD of 𝒟*_s_* and 𝒟*_t_* after the feature extractor ***G****_f_* can be expressed as:(3)$$L_{MMD}\left(D_{s},D_{t}\right)=\underset{G_{f}\in ℱ}{\sup}\;\left(E_{D_{s}}[G_{f}(x_{s})]-E_{D_{t}}[G_{f}(x_{t})]\right)$$
where ℱ represents the reproducing kernel Hilbert space (RKHS). sup(·) is the supremum of the input. By replacing the population expectations with empirical expectations, a biased empirical estimate of MMD can be obtained and written as:(4)$$L_{MMD}\left(D_{s},D_{t}\right)=\underset{G_{f}\in ℱ}{\sup}\;\left(\frac{1}{m}\sum_{i=1}^{m}G_{f}(x_{s}^{i})-\frac{1}{n}\sum_{j=1}^{n}G_{f}(x_{t}^{j})\right)$$

By means of the kernel mean embedding of distribution, RKHS is induced by the characteristic kernels, such as Gaussian and Laplace kernels [38]. The empirical estimate of the squared MMD is defined as:(5)$$L_{MMD}{}^{2}\left(D_{s},D_{t}\right)=\frac{1}{m^{2}}\overset{m}{\underset{i,j=1}{\sum }}k\left(x_{s}^{i},x_{s}^{j}\right)-\frac{2}{mn}\overset{m,n}{\underset{i,j=1}{\sum }}k\left(x_{s}^{i},x_{t}^{j}\right)+\frac{1}{n^{2}}\overset{n}{\underset{i,j=1}{\sum }}k\left(x_{t}^{i},x_{t}^{j}\right)$$
where *k*(.,.) is the characteristic kernel. In order to avoid the difficulty of selecting the kernel function, MK-MMD assumes that the optimal kernel can be obtained linearly from multiple kernels. The characteristic kernel associated with the feature map *f*, *k*($x_{s}^{i}$, $x_{t}^{j}$) = <*f*($x_{s}^{i}$), *f*($x_{t}^{j}$)>, is defined as the convex combination of *N* kernels {*k_u_*}:(6)$$K≜\left\{k=\sum_{u=1}^{N}\beta_{u}k_{u}:\sum_{u=1}^{N}\beta_{u}=1,\beta_{u}\geq 0,\forall u\right\}$$
where {*β_u_*} are the constraints on coefficients and are imposed to guarantee that the derived multi-kernel *k* is characteristic [39]. Multiple Gaussian kernels with different radial basis function (RBF) bandwidths are widely used as a nonparametric method.

In the forward-training process, the training of ***G****_f_* can make the features discriminative. However, in order to make the extracted features domain-invariant, ***G****_f_* should furthermore maximize the domain classification loss, which is run in the opposite direction to the training of ***G****_d_*. In order to implement the adversarial training between ***G****_f_* and ***G****_d_*, GRL is inserted between the feature extractor and the domain classifier. In the forward propagation-based training, GRL acts as an identity transformer. However, during the back propagation-based training, the GRL multiplies the gradient by a certain negative constant −*λ*, leading the domain classification loss negative feedback to ***G****_f_*. A detailed discussion of GRL can be found in Ref. [30].

Therefore, by incorporating the MK-MMD loss into the adversarial training, the discriminative and domain-invariant features can be learned by the feature extractor. The total loss function of the feature extractor ***G****_f_* is expressed as:(7)$$L_{f}\left(D_{s},D_{t}\right)=L_{y}\left(D_{s}\right)-\lambda L_{d}\left(D_{s},D_{t}\right)+\alpha L_{MMD}\left(D_{s},D_{t}\right)$$
where α is the trade-off hyperparameter of MK-MMD loss.

Based on the above loss function of ***G****_f_*, ***G****_y_*, and ***G****_d_*, the training is performed. The corresponding parameters *θ_f_*, *θ_y_*, *θ_d_* of ***G****_f_*, ***G****_y_*, and ***G****_d_* are updated as follows:(8)$$\theta_{f}\leftarrow \theta_{f}-\mu \frac{\partial L_{f}}{\partial \theta_{f}}$$
(9)$$\theta_{y}\leftarrow \theta_{y}-\mu \frac{\partial L_{y}}{\partial \theta_{y}}$$
(10)$$\theta_{d}\leftarrow \theta_{d}-\mu \frac{\partial L_{d}}{\partial \theta_{d}}$$
where *μ* is the learning rate. In this paper, the stochastic gradient descent (SGD) algorithm with 0.9 momentum and an annealing learning rate is used to optimize the model parameters [30]. The pseudo code of the proposed MMD-DANN model is presented in Appendix A.

### 2.3. Detection Procedure of Target Domain

When the training is complete, the unlabeled target domain data can be classified using the trained feature extractor and the trained label predictor. The target data are first input into the feature extractor to extract features vector, then fed forward into the label predictor, and the sigmoid activation function predicts the class labels. The detection procedure of the proposed MMD-DANN model is presented in Figure 2 and can be described as follows:

First, the Lamb wave response signals of center-hole metal structures are obtained by FEM simulations and fatigue tests. The simulated response signals under different conditions are considered as the labeled source domain data, and the experimental response signals are considered as the unlabeled target domain data.

Then, in the step of model training, the labeled source domain data and unlabeled target domain data are fed forward into the MMD-DANN model. The extracted features vector, the predicted class labels, and the predicted domain labels are obtained. Three loss functions are calculated and propagated backward to update the model parameters until the training of the proposed MMD-DANN model is finished.

Finally, the unlabeled target domain data are used as testing samples to input to the trained model, and the damage detection results are obtained.

## 3. Experimental Validation

The proposed unsupervised domain adaptation method is validated on the fatigue crack detections of a center-hole metal structure. We aimed to detect the fatigue crack of the metal structures by transferring the learning knowledge from the abundant labeled source domain into an unlabeled target domain, in which the source domain data are assumed as the simulated response signals under different conditions and the target domain data are assumed as the experimental response signals under uncertain conditions. The simulated and experimental dataset are used to demonstrate the transfer results of the proposed MMD-DANN model and the detection accuracy of the experimental dataset.

### 3.1. Simulated Dataset and Data Preprocessing

The dataset consists of Lamb wave response signals of metal structures manufactured using 7050 aluminum of 3 mm thickness with a center-hole 25 mm diameter from simulation and experiment. Material properties of the center-hole metal specimen are shown in Table 1. Four PZT sensors P51 were installed on every specimen to monitor the healthy conditions of both sides of the center-hole. Two sensing paths, A1-S1 and A2-S2, were formed, in which A1 and A2 serve as actuators and S1 and S2 serve as sensors. The dimensions of PZT sensors are 8 mm in diameter and 0.45 mm in thickness. In order to demonstrate the detection performance on small fatigue cracks of the proposed method, only fatigue cracks under 8 mm are studied. Specimen geometry and PZT sensors placement are shown in Figure 3. The material and structure form considered as specimen are common in the aircraft. The FE model and the physical fatigue test are built with the same specimen geometrical dimensions and PZT sensors placement.

Considering the symmetries of structure geometry and sensors placement, FEM simulations are performed to acquire the response Lamb wave signals of only A1-S1 sensing path. The PZT sensor is modeled by an electromechanical coupling plate element, and the positive and negative piezo-conductive effects of actuators and sensors can be realized [41]. In order to simulate the uncertainty of the actual crack morphology, fatigue crack is modeled with a notch of 0.05 mm width, different length, and different orientations. Hanning window tone burst signals with a five-cycle frequency of 230 kHz is used as excitation signals. The central frequency is referred to the experiment. A fixed time increment of 0.1 μs is consistent with sampling rate of the experiment, which is sufficient to capture the interested time period of signals. The output time of field results for sensors is set as 4 ms to ensure that the number of the output data point is consistent with the experiment. The global element size is 1 mm, and the local encrypted element size of crack and hole are 0.5 mm.

The availability of the FE model is validated by comparing the simulated group velocity and the analytical group velocity for the same frequency. The simulated group velocity of S_0_ mode is estimated as 5381 m/s, and the simulated group velocity of A_0_ mode is estimated as 2955 m/s. The respective errors for group velocity of S_0_ and A_0_ mode are about 1.93% and 2.96%. In this paper, the response signals of S_0_ mode are used to analyze. A two-dimensional FE model of the center-hole specimen with 5 mm fatigue crack and 90° orientation is created in ABAQUS/Explicit with one edge fixed and another edge loaded with 3 kN, as shown in Figure 4. The holding load is the same as the experiment. To illustrate the effect of crack on the wave propagation process, a screenshot of the simulated wave propagation of the center-hole specimen with 5 mm fatigue crack and 90° orientation is shown in Figure 5. The displacement at 85 μs can indicates the wave reflections by the damage and the top boundary, which further demonstrates the availability of the simulations.

Simulation was performed for an undamaged case followed by multiple damaged cases. Damaged cases are induced at sixteen different crack lengths (from 0.5 mm to 8 mm at a step of 0.5 mm) with five different orientations (80°, 85°, 90°, 95°, 100°). Simulation was performed for one damage case at a time. Considering that direct wave packet contains the most effective structural information in the response signals, a fixed-length rectangular window [60 μs 105 μs] was employed to extract a direct wave packet for S_0_ mode. Consequently, 81 simulated samples were 80 samples from damaged cases and the remaining 1 sample is undamaged, with each sample consisting of 451 data points which corresponded to the time instances of the extracted direct wave packet. Figure 6 shows the simulated Lamb wave response signals of different crack lengths for the A1-S1 sensing path. Significant phase right-shifting and amplitude decreasing can be observed with the undamaged and damaged conditions, which means that the simulated signals can reflect the variations of structural conditions.

In order to consider the effect of dispersions of specimens and sensors performance on the signals, data augment technology was used to introduce the fluctuations of amplitude and phase into the signals to expand the simulated samples. Firstly, the signals are converted into the analytical signals with Hilbert transform. Then, the amplitude and phase of the analytical signals are multiplied by a scaling coefficient to generate the virtual simulated samples. For the undamaged sample, the scaling coefficient for the amplitude varies from 0.82 to 1.20 at a step of 0.02, and the scaling coefficient for the phase varies from 0.82 to 1.21 at a step of 0.03. The scaling of the amplitude and phase are cyclic preceded. For the damaged samples, the scaling coefficient for the amplitude varies from 0.80 to 1.25 at a step of 0.15, and the scaling coefficient for the phase varies from 0.80 to 1.25 at a step of 0.15. The scaling of the amplitude and phase are meanwhile preceded. That is, a total of 600 simulated samples are obtained with 280 undamaged samples and 320 damaged samples.

### 3.2. Experimental Dataset and Data Preprocessing

Fatigue tests on eight center-hole metal specimens were performed to obtain the experimental response signals, labeled from T1–T8. The experimental setup is shown in Figure 7. For a constant amplitude axial tensile cyclic load with load ratio *R* = 0.1, the maximum load of 40 kN and loading frequency of 8 Hz was applied to the specimens using a hydraulic MTS machine to introduce fatigue cracks in the specimens. The initiation and growth of fatigue cracks were measured offline with a charge-coupled device (CCD) camera. Crack lengths are measured from optical microscopic images of CCD camera. Representative examples of actual fatigue crack through the microscopic are shown in Figure 8. At the early stages of the fatigue test, only one crack initialized and grew. At the medium stage of the fatigue test, two cracks grew simultaneously.

An integrated structural health monitoring system was utilized to generate and acquire online Lamb waves. During the fatigue test, Lamb waves were periodically acquired when the load was held at 3 kN. The load holding was to ensure the identical boundary condition for every signal acquisition. The temperature variation at the laboratory was maintained below 1 °C to minimize temperature effects on acquired Lamb waves [42]. The excitation signal is a five-cycle tone burst modulated by a Hanning window with the sampling rate set to 10 MHz and the sampling length set to 4000.

By analyzing the waveform envelope and the arrival wave packet overlap under 150 kHz: 20 kHz: 270 kHz excitation frequencies, typical response signals of A1-S1 sensing path under 230 kHz are given in Figure 9. The separated first arrival wave packet can be observed without overlapping with the crosstalk signals and boundary refection signals, and the experimental group velocity of S_0_ mode is approximately estimated as 5106 m/s. Therefore, the central frequency of excitation signal was confirmed as 230 kHz to obtain separated direct wave packet. A fixed-length rectangular window [82 μs 127 μs] was employed to extract the direct wave packet. The non-coupling of A1-S1 and A2-S2 sensing paths is demonstrated by comparing the response signals of A2-S2 sensing path only with one side crack and that with two sides cracks. Thus, the response signals of each sensing path are regarded as one independent sample. Figure 10 plots the experimental Lamb wave response signals of different length for specimens T1–T3. Due to the dispersions of specimens, sensor performance, and installation process, the response signals amplitude and damage sensitivities for different specimens are different, and are manifested as different changes in amplitude and phases under the same crack length growth. Further comparison of the simulated and experimental response signals shows that the direct wave packet can be extracted earlier in simulated data, and the simulated data have higher damage sensitivity.

The experimental dataset of eight center-hole metal specimens consists of one hundred and forty undamaged samples and eighty damaged samples, with each sample consisting of 451 data points, in which the undamaged and damaged samples are extremely unbalanced. In order to balance the undamaged and damaged samples to further improve the model performance, data augment technology is also applied to introduce the fluctuations of amplitude and phase into the response signals to expand the experimental samples. For the undamaged samples, the scaling coefficient for the amplitude varies at (0.90, 1.20), and the scaling coefficient for the phase varies at (0.90, 1.20). Meanwhile, the scaling of the amplitude and phase are preceded. For the damaged samples, the scaling coefficient for the amplitude varies from 0.80 to 1.25 at a step of 0.15, and the scaling coefficient for the phase varies from 0.80 to 1.25 at a step of 0.15. The scaling of the amplitude and phase are meanwhile preceded. That is, a total of 600 experimental samples are obtained with 280 undamaged samples and 320 damaged samples.

Descriptions of the simulated and experimental dataset are shown in Table 2. The dataset is randomly divided as training and testing datasets at a ratio of 4:3. Out of 600 cases of simulated data and experimental data, 210 undamaged cases and 240 damaged cases are used for training while 70 undamaged cases and 80 damage cases are used for testing. In addition, max-min normalization is a necessary step to convert the different scales of the simulated and experimental dataset into a common scale, which enables the unbiased contribution from the output of every response signal.

### 3.3. Transfer Results of MMD-DANN Model

In the proposed MMD-DANN model, *λ*, and *α* are important trade-off parameters which seriously affect the transfer performance of the model. Thus, the studies of the trade-off parameters are implemented first. The classification accuracy is related to distribution divergence between two domains. Thus, MMD of the extracted features output by the second fully connected layer of the label predictor is used to evaluate the transfer performance with different trade-off parameters, which is the highest-level feature before classification.

In order to suppress noisy signals from the domain classifier at the early stages of the training procedure [31], the parameter *λ* gradually changed from 0 to 1 using the formula 2/(1 + e^−10*p*^) − 1 instead of a fixed value, in which *p* is the training progress linearly changing from 0 to 1. With the training process progressing, the trade-off parameter *λ* gradually increases. The parameter *α* is selected from {0.01, 0.05, 0.1, 0.5, 1, 5, 10, 50}. In order to improve the optimization of SGD during the training, an annealing learning rate is adopted using the formula *μ* = *η*_0_/(1 + 10*p*)^0.75^, in which *η*_0_ is the initialized learning rate. The SGD method with an initialized learning rate 1 × 10^−3^ and mini-batch 64 was used to train the model. Every training is carried out for five trails and the average us obtained to reduce the effect of randomness. All training is performed using python on the Inter(R) Xeon(R) Gold 6462R CPU.

Figure 11a shows MMD of the extracted features with different trade-off parameter α. The classification accuracy of different parameter *α* on the experimental dataset is given in Figure 11b. It can be seen that MMD of the extracted features and the corresponding classification accuracy have a nonmonotonic trend with the parameter *α*, but the smaller MMD corresponds to the higher classification accuracy. MMD takes the minimum value when *α* is set as 5 and 50, and the classification accuracy takes the maximum value when *α* is set as 5, and the corresponding classification accuracy of the experimental dataset is 98.12%. Therefore, MMD-DANN model with the trade-off parameter *α* of 5 is trained to detect fatigue cracks.

To further demonstrate the superiority of the progressing strategy for trade-off parameter *λ*, the transfer results of the model with a fixed parameter *λ* are compared. The parameter *λ* is selected from {0.1, 0.5, 1, 5, 10}. Figure 12 shows MMD of the extracted features and classification accuracy with different parameter *λ* when *α* is set to 5. The parameter *λ* has an obvious effect on the transfer results of the model. Once the parameter *λ* is set as a large value, such as being larger than 1, the classification accuracy of the model sharply drops to a poor level. A safe way is to select a small parameter *λ* to balance the domain classification loss and the domain adaptation loss in the loss function. Especially, the classification accuracy takes the maximum value when *λ* is set as 1 and *α* is set as 5, and the corresponding classification accuracy of the experimental dataset is 94.33%, which is still lower than when *λ* is set as a changing value and *α* is set as 5. The progressive training strategy of the trade-off parameter *λ* significantly improves the classification performance and simplifies the parameter-selecting. Finally, the MMD-DANN model, with a changing trade-off parameter *λ* and a fixed trade-off parameter *α* of 5, was trained to detect fatigue cracks.

### 3.4. Comparisons with Other Methods

To further validate the transfer results and transfer performances of the proposed method, we compared our method with other methods, including 1D-CNN, WGAN-GP, DDC, DAN, and DANN.

For comparison, 1D-CNN consists of a feature extractor and a label predictor, in which the architectures of the feature extractor and the label predictor are the same as MMD-DANN model. A 1D-CNN model trained on the labeled simulated data was used to classify the unlabeled domain data without domain adaptation knowledge. The architecture of CNN classifier in WGAN-GP model is the same as the 1D-CNN model. WGAN-GP is a classical adversarial generation-based deep domain adaptation method made up of a generator, a discriminator, and the gradient penalty. By the adversarial training between the generator and the discriminator, a CNN classifier trained on the synthetic domain can be directly transferred to the target domain. The architecture of CNN classifier in WGAN-GP model is the same as a 1D-CNN model. DDC is a common unsupervised domain adaptation method made up of a fixed CNN, an adaptation layer, and MMD, whose architecture analogous MMD-DANN model is without a domain classifier. The position to place the adaptation layer in is decided by comparing every condition. By introducing more adaptation layers and MK-MMD, DAN is developed, which is made up of a feature extractor, a label predictor, and MK-MMD. The trade-off parameter *α* of MK-MMD in the loss function for DAN model is searched from {0.1, 0.5, 1, 5, 10, 50}. The optimal transfer result of DAN model is calculated as 0.5. DANN model is the simplified version of MMD-DANN model, without MK-MMD in the loss function. The training dataset of 1D-CNN, WGAN-GP, DDC, DAN, DANN, and the proposed model are 70% of the labeled simulated domain data and 70% of the unlabeled target domain data. The testing dataset of 1D-CNN, WGAN-GP, DDC, DAN, DANN, and the proposed model are the experimental domain data. Every model carried out six trainings. The network architecture of 1D-CNN, WGAN-GP, DDC, DAN, and DANN with all hyperparameters used in the paper is shown in Appendix B.

To observe the transfer results specifically, Table 3 denotes MMD for six models, in which MMD for CNN model is calculated from the source and target data, equal to initial MMD without domain adaptation. Based on the domain adaptation mechanisms of other five models, the transfer result for WGAN-GP model can be expressed as MMD of the synthetic and target data, and the transfer results for DDC, DAN, DANN, and the proposed model can be expressed as MMD of the extracted features vector output by the feature extractor. Compared with six models, the MMD of raw data is the highest, and the MMD of five deep domain adaptation methods is smaller, indicating that the deep domain adaptation methods are effective tools to reduce the distribution discrepancy. The proposed method has the smallest MMD, implying that the proposed method has the best learning ability of domain adaptation.

To evaluate the performance of models comprehensively, false alarm rate and missing alarm rate were introduced to enrich the evaluation of the classification accuracy of the proposed MMD-DANN and other methods. The false alarm rate is defined as the proportion of undamaged samples predicted as damaged samples in total predicted damaged samples. The missing alarm rate is defined as the proportion of damaged samples predicted as undamaged samples in total damaged samples.

The average classification accuracy, false alarm rate, and missing alarm rate on the experimental dataset are detailed in Table 4. The classification accuracy of five trails in every method is shown in Figure 13. The average classification accuracy of the proposed MMD-DANN model on the experimental dataset is 98.12%, which is the highest among six methods. The average false alarm rate and missing alarm rate of the proposed MMD-DANN model are 3.08% and 1.56%, which are also the smallest among the six methods.

From the results shown in Table 4, due to the lack of domain adaptation procedure, the average classification accuracy of 1D-CNN model is just 58.53%, which is the smallest among all the methods. The average false alarm rate and missing alarm rate of 1D-CNN model are 30.69% and 33.33%, which are the highest among six methods. The detection results of 1D-CNN model further demonstrate that the mismatched distributions of the source and target domain data cause reduced performance in the target domain.

As shown in Table 4, the average classification accuracy of WGAN-GP model is 66.93%, which is the smallest in five deep domain adaptation methods but only higher than 1D-CNN. The average false alarm rate and missing alarm rate of WGAN-GP model are 23.88% and 30.28%, which is the highest among five deep domain adaptation methods. Due to the dispersions of specimens, sensor performance, and the installation process, the experimental signals of different specimens still have an obvious distinction (as shown in Figure 10), which deteriorates the synthetic data quality. Therefore, the large distribution discrepancy between the synthetic data and the target data leads the classifier trained on the synthetic data to show a poor classification performance on the target data. It can be inferred that the complexity of the target domain data has an important effect on the adversarial generation results of WGAN-GP model. Further, by comparing the detection results of 1D-CNN with WGAN-GP model, the classifier of WGAN-GP model presents a better classification performance because of the smaller distribution discrepancy between the synthetic and target data compared to that between the simulated and experimental data.

According to the detection results in Table 4, the average classification accuracy of DDC model and DAN model are 75.62% and 80.00%, which identifies the better domain adaptation performance of multiple adaptation layers and MK-MMD for the DAN model than that of the single adaptation layer and MMD for the DDC model. Based on the comparison results in Table 4, the average classification accuracy of DAN model and DANN model are 80.00% and 84.73%, which are close to each other, meaning that for the fatigue crack detection scenarios, the domain adaptation performance of minimizing MMD is equal to the adversarial training. The average classification accuracy of DAN model and DANN model are higher than WGAN-GP model, because the distribution discrepancy between the high-dimensional features vector for DAN and DANN model are much smaller than that between the synthetic and target data. Specially, the average classification accuracy of DAN and DANN model are smaller than MMD-DANN model, because DAN model reduces the distribution discrepancy just by minimizing MMD and DANN model reduces the distribution discrepancy just by adversarial training, but MMD-DANN model reduces the distribution discrepancy by combing MMD and adversarial training. The comparison results further verify the effectiveness of domain adaptation and adversarial training of the proposed MMD-DANN model.

A confusion matrix for the first training trail of the proposed model on the experimental dataset is shown in Figure 14. For 280 undamaged and 320 damaged experimental data, 3.57% of undamaged samples were misclassified as damaged conditions. All damaged samples are all classified accurately. It can be inferred that the proposed model can accurately classify undamaged and damaged conditions.

In order to visually validate the transfer learning effectiveness of the proposed method, the t-distributed stochastic neighbor embedding (t-SNE) [43] technique was used to map the high-dimensional features vector into a two-dimensional space. The mapped transferrable features for six models are shown in Figure 15. From the result shown in Figure 15, the transferable learning features of the undamaged source data have an obvious gathering cluster because the undamaged source data are obtained with one undamaged model, and the transferable learning features of the damaged source data have multiple gathering clusters because of the damaged source data corresponding to different crack lengths. For the experimental target data, the undamaged and damaged data are scatter-distributed because of the dispersions among specimens. 

From Figure 15a, the transferable features extracted by 1D-CNN model suffer from poor distribution discrepancy. With the dash line plotted in Figure 15a, the source data can be accurately classified, while many undamaged and damaged target domain data are misclassified. Thus, 1D-CNN model trained on the source data cannot accurately classify the unlabeled target data. From Figure 15b, the transferable features of the target domain extracted by WGAN-GP model have a small among-class distance. The transferable features cannot be effectively obtained using the WGAN-GP model. In addition, compared with 1D-CNN model, with the dash line plotted in Figure 15b, less damaged source data are misclassified. Consequently, the classification accuracy of WGAN-GP model is higher than the 1D-CNN model. From Figure 15c–e, the transferable features learned by DAC, DAN, and DANN model have a smaller cross-domain discrepancy. As a result, DAC, DAN, and DANN models showed higher classification accuracy on the unlabeled target data than 1D-CNN and WGAN-GP models. However, with the dash line plotted in Figure 15c–e, some undamaged samples are falsely alerted, and some damaged experimental samples are missing alerted. As shown in Figure 15f, the transferable features of two domains under the same class are projected into the same region, and the different conditions data are separated well. The result shows that the proposed MMD-DANN model not only effectively reduces the inter-domain distance but also enlarges the among-class distance. With the dash line plotted in Figure 15f, only a few damaged target data are misclassified. The corresponding crack length of the misclassified damaged target samples are all 3 mm, because the response signals with 3 mm crack length have a small variation compared to the undamaged signals. Consequently, the classification accuracy of the proposed model is the highest. The visualization results prove that the proposed MMD-DANN model outperforms the other methods and presents the best transfer performance. The proposed MMD-DANN model can be used as an effective domain adaptation tool to accurately detect fatigue crack for the unlabeled experimental samples.

## 4. Conclusions

In this paper, an automated fatigue crack detection method based on MMD-DANN model was proposed to accurately detect structural conditions for metal structures. To overcome the difficulty of time-consuming acquisitions of the labeled data in practice, FEM simulations were adopted to obtain simulated response Lamb wave signals with different healthy conditions, which are assigned as the labeled source domain data. Due to the distribution discrepancy between the simulated and experimental signals, the classifier model trained on the simulated data cannot be directly applicable to the experimental data. A novel unsupervised domain adaptation method based on MMD-DANN model was developed. By integrating MMD with the adversarial training of DANN model, the discriminative and domain-invariant features of the simulated source domain and the experimental target domain can be extracted. Following that, the classification knowledge of the labeled source domain can be generalized to the unlabeled target domain. Fatigue tests on center-hole metal specimens are implemented to validate the proposed method. Comparing with the data-driven intelligent method and other deep domain adaptation methods, the detection results on the experimental data show that the proposed MMD-DANN model presents higher classification accuracy and better transfer performance. The average classification accuracy on the experimental data for MMD-DANN model is 98.12%, and the false alarm rate and missing alarm rate are 3.08% and 1.56%, showing the domain adaptation effectiveness of the proposed MMD-DANN model.

The proposed method is a regular binary classification method which only classifies the undamaged and damaged conditions, and is unable to further classify damaged cases with different crack length. Therefore, in our future research, an automated damage quantification detection method for fatigue crack based on unsupervised deep domain adaptation will be studied. Moreover, unsupervised domain adaptation detection methods for complex multi-bolt joint specimens and lap specimens also need to be established.

## Figures and Tables

**Figure 1 sensors-23-01943-f001:**
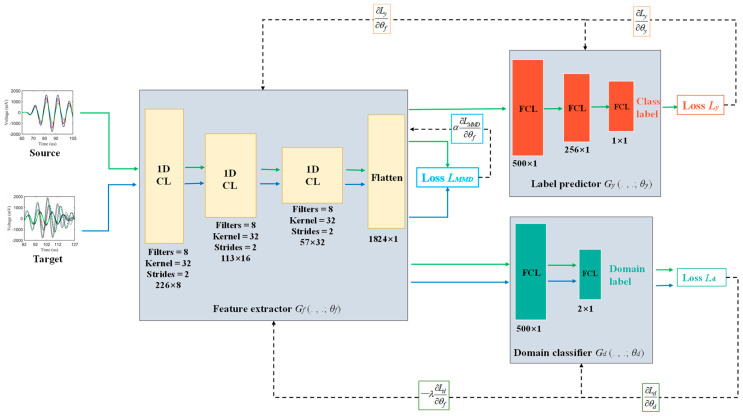
The architecture of the proposed MMD-DNN model.

**Figure 2 sensors-23-01943-f002:**
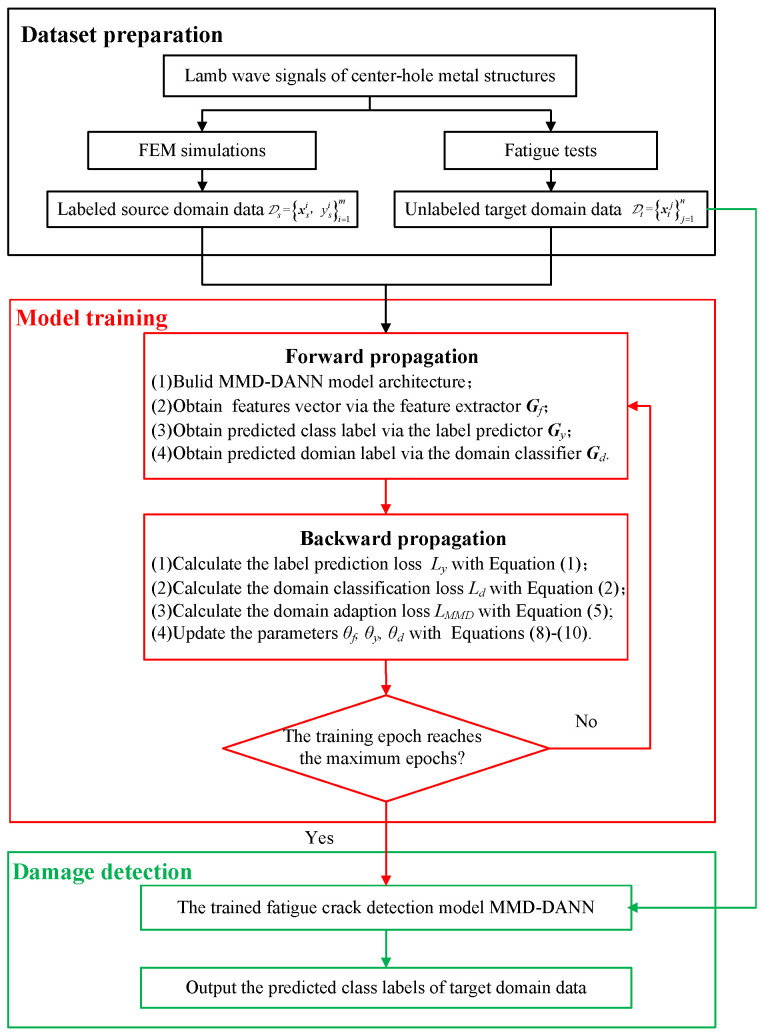
Flowchart of the proposed fatigue crack detection method based on MMD-DANN model.

**Figure 3 sensors-23-01943-f003:**
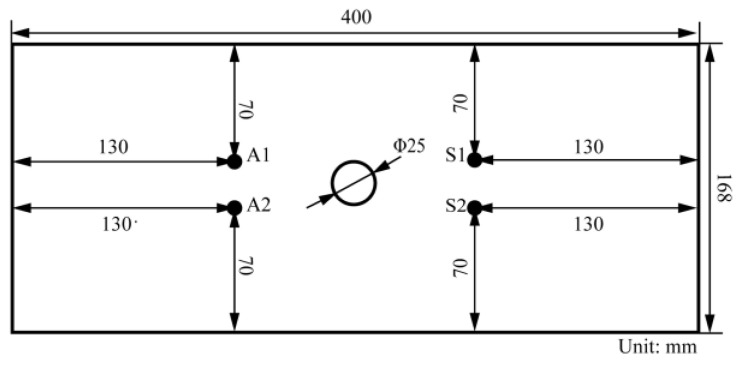
Specimen geometry and PZT sensors placement.

**Figure 4 sensors-23-01943-f004:**
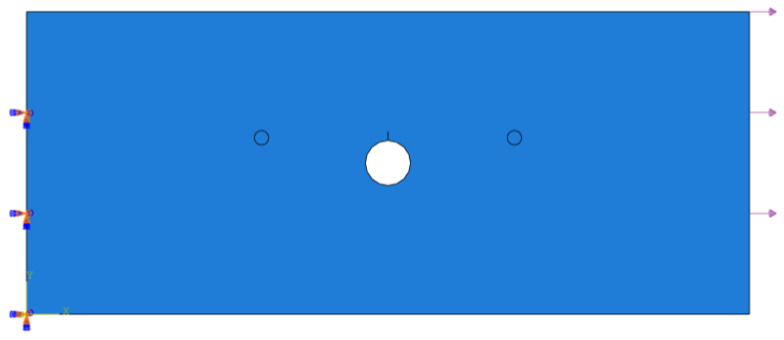
FE model of center-hole metal specimen using ABAQUS/Explicit.

**Figure 5 sensors-23-01943-f005:**
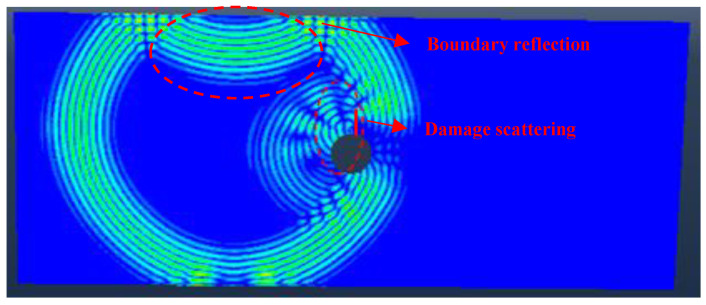
Total displacement nephogram of simulated wave propagation at 85 μs.

**Figure 6 sensors-23-01943-f006:**
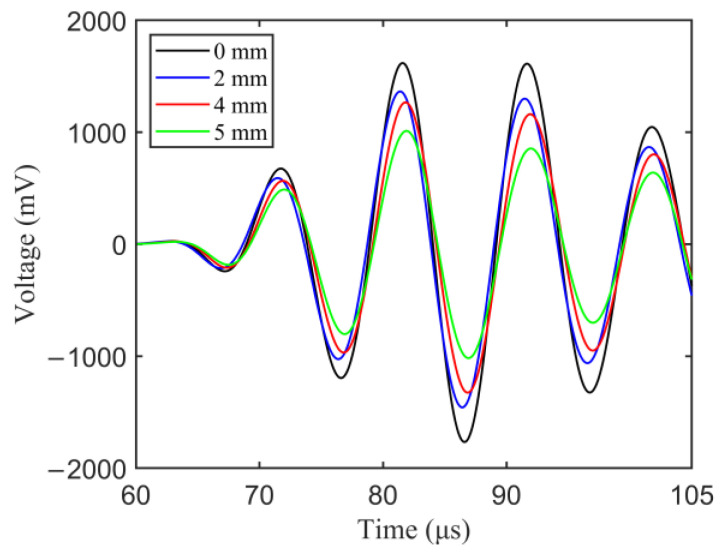
Simulated Lamb wave response signals of different crack length.

**Figure 7 sensors-23-01943-f007:**
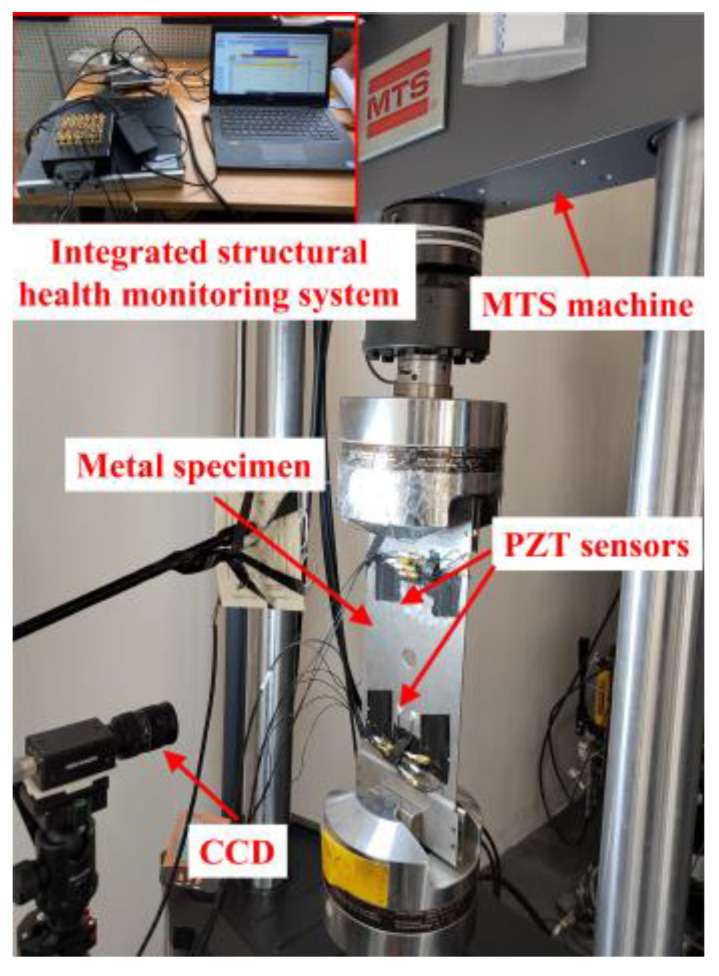
Experimental setup for center-hole metal specimens.

**Figure 8 sensors-23-01943-f008:**
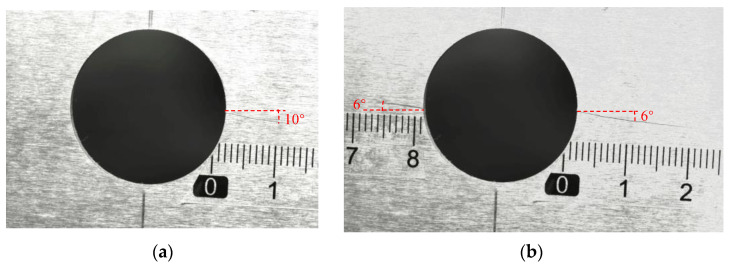
Cracks observed from the microscope of specimen T1 at: (**a**) 59,673 load cycles and (**b**) 70,766 load cycles.

**Figure 9 sensors-23-01943-f009:**
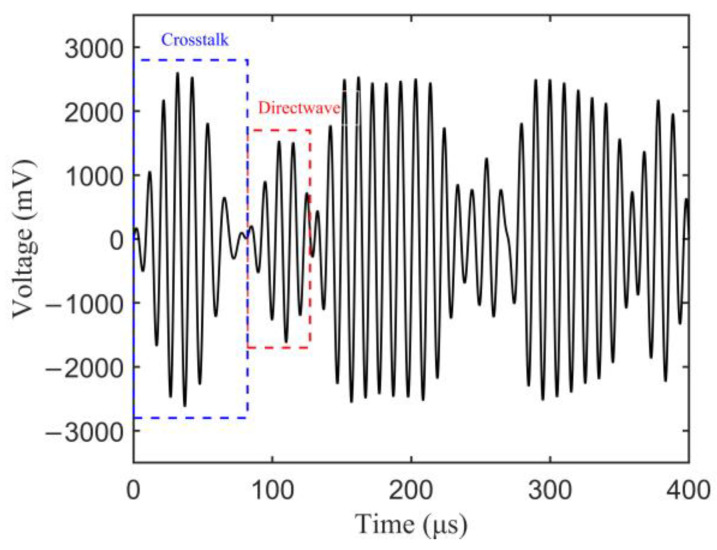
Typical response signals of A1-S1 sensing path under 230 kHz excitation frequency.

**Figure 10 sensors-23-01943-f010:**
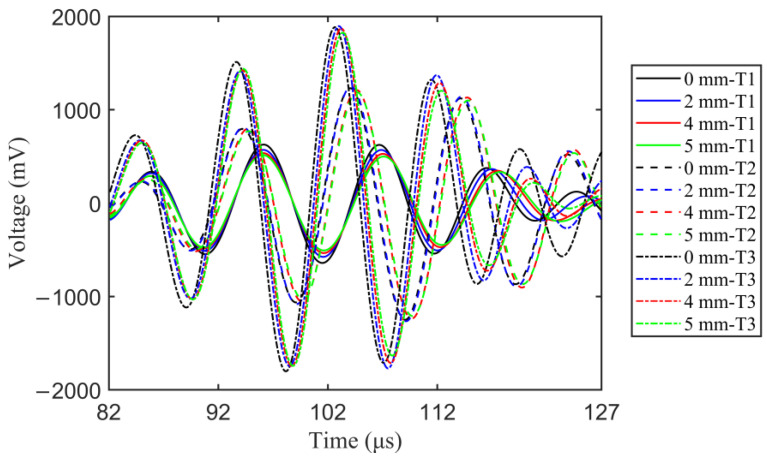
Experimental Lamb wave response signals of different crack lengths for specimens T1–T3.

**Figure 11 sensors-23-01943-f011:**
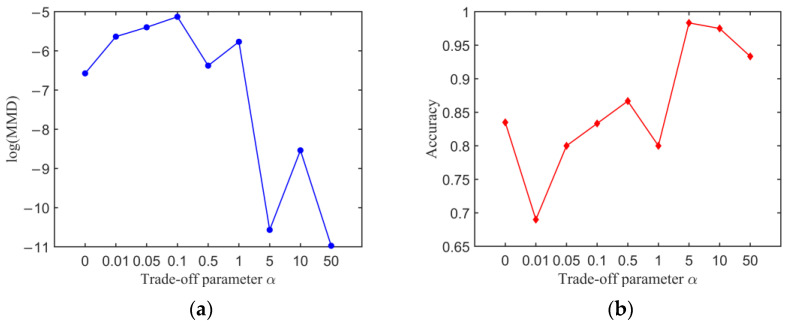
Transfer results of MMD-DANN model with different trade-off parameter *α*: (**a**) MMD of the extracted features and (**b**) classification accuracy.

**Figure 12 sensors-23-01943-f012:**
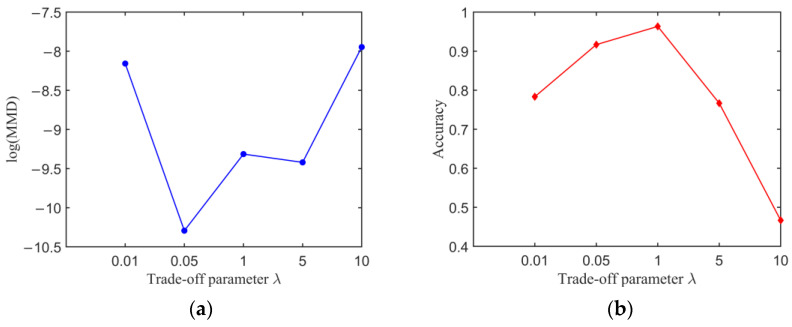
Transfer results of MMD-DANN model with different trade-off parameter *λ*: (**a**) MMD of the extracted features and (**b**) classification accuracy.

**Figure 13 sensors-23-01943-f013:**
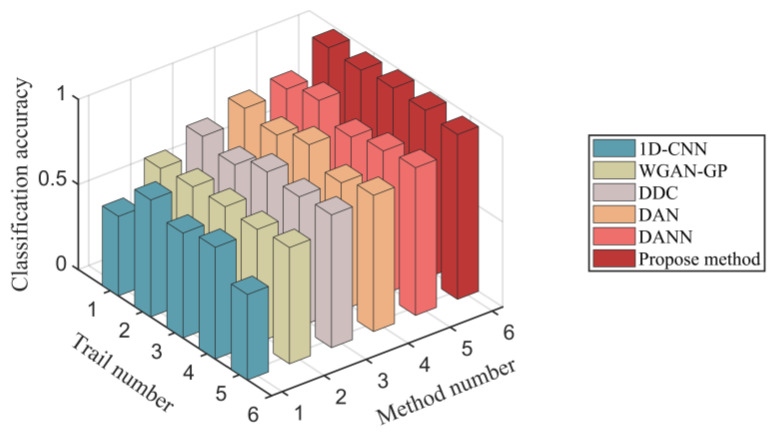
The classification accuracy comparison of multiple methods.

**Figure 14 sensors-23-01943-f014:**
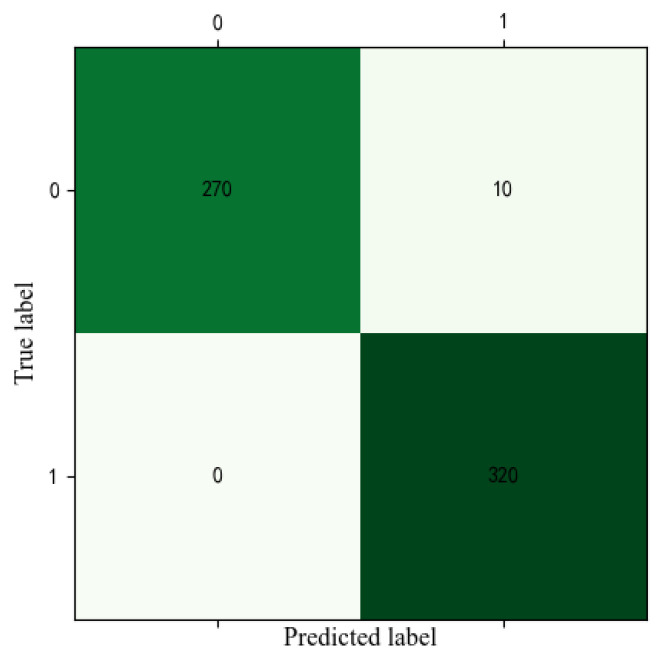
The confusion matrix of the proposed method on the experimental dataset.

**Figure 15 sensors-23-01943-f015:**
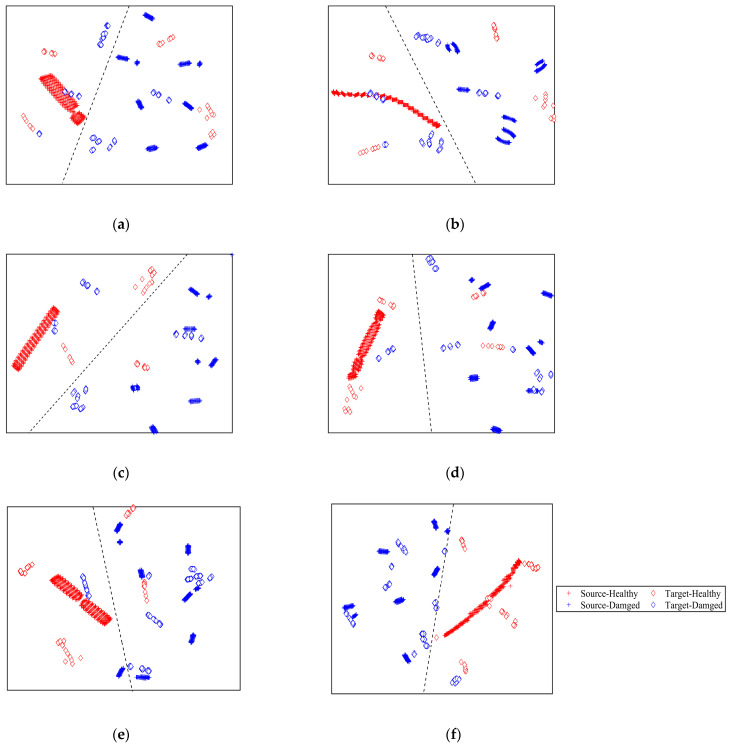
The visualization of the mapped features on the source and target domain: (**a**) 1D-CNN, (**b**) WGAN-GP, (**c**) DDC, (**d**) DAN, (**e**) DANN, and (**f**) MMD-DANN.

**Table 1 sensors-23-01943-t001:** Material properties of center-hole metal specimens.

*ρ*/(kg/m^3^)	*E*/GPa	*ν*
2700	70	0.33

**Table 2 sensors-23-01943-t002:** Introduction to datasets.

Dataset	Healthy Conditions	Number of Samples
Simulation	Undamaged	280
	Damaged	320
Experiment	Undamaged	320
	Damaged	280

**Table 3 sensors-23-01943-t003:** Transfer results of six models.

Models	Data	MMD
1D-CNN	Raw source data and target data	0.008052
WGAN-GP	Synthetic data and target data	0.003773
DDC	Extracted features vector of source data and target data	0.003083
DAN	Extracted features vector of source data and target data	0.002932
DANN	Extracted features vector of source data and target data	0.001397
Proposed	Extracted features vector of source data and target data	0.0001998

**Table 4 sensors-23-01943-t004:** Detection results for experimental dataset with multiple methods.

Methods	Accuracy (%)	False Alarm Rate (%)	Missing Alarm Rate (%)
1D-CNN	58.53	30.69	33.33
WGAN-GP	66.93	23.88	30.28
DDC	75.62	28.34	15.87
DAN	80.00	26.83	6.25
DANN	84.73	15.14	13.96
Proposed	98.12	3.08	1.56

## Data Availability

Not applicable.

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
