# Peer review of "FEM Simulation-Based Adversarial Domain Adaptation for Fatigue Crack Detection Using Lamb Wave"

_sensors, 2023, doi:10.3390/s23041943_

Round 1
Reviewer 1 Report
The paper proposed a novel method based on FEM simulation and adversarial domain adpation method for Lamb wave analysis for fatigue crack detection. The idea is interesting and significant. The paper needs to answer the following questions to improve and correct.
1. The motivation for the paper needs to be empysized and made to much clear. The FEM data was used to solve the difficulties in data shortage for experimental data. Do the FEM data come from the FEM model with correct simuliation? What is the requirement for FEM? What is the physical relation between the simulated data and experimental signal?
2.What is the advantage for adversarial domain adaptition method for special role in the idea of the paper?
3. The FEM simuation data is obtained by FEM model with special parameters, which means that there is one simulated signal under one model for one crack. What is the number of training data? The paper should add the procedure about the proposed method.
4. The structure of the paper should be improved. The section 2 is about the introduction to unsupervised deep domain adaption, which need to be included in section 1 and section 3.
5. In Figure 2 , there is error in backward propagation about the parameters.
6. The results in Figures6 and10 are confused and need to be check for correction.
7. In introduction, the references analyzed the existed method in current papers, which explain the background for the proposed method. The current novel methods for other researchers should be analyzed and described. For example: On line39-41 , "hower ........their threshold to identify the structural damages". This results are not exactly the truth and correct. Pls analyzed bycomparision with refs "Model-based method with nonlinear ultrasonic system identification for mechanical structural health assessment" "Intelligent early structural health prognosis with nonlinear system identification for RFID signal analysis"
8 The algorithms about the proposed MMD-DANN is not very clear and should be further described.
9 The english should be improved.
Reviewer 2 Report
This manuscript is about the detection of fatigue cracks utilizing Lamb waves and the finite element method (FEM) with adversarial domain. Congratulations to the authors for this job. The topic might be of interest to an audience of the journal. However, the reviewer has the following major concerns:
1. The authors need to make the contribution clear and support with enough motivation. The current state of the manuscript does not provide much motivation and novel contributions from the authors.
2. There are more relevant references available for the damage detection based on deep learning, e.g., Sampath, S., Jang, J., & Sohn, H. (2022). To strengthen the literature review on deep learning based damage detection, I suggest to cite this and discuss these in the introduction section.
3. It would be better to add the implementation code or pseudo code in the appendix section.
4. The reviewer suggests considering several existing deep-learning models (at least five models) in order to validate the proposed deep-learning model
Round 2
Reviewer 2 Report
The authors have adequately addressed the comments made by the reviewers in the revised version of the manuscript. Therefore, I have no further comments